# Urbanicity, biological stress system functioning and mental health in adolescents

**Brittany E. Evans** [1,2]*, **Anja C. Huizink**[3,4], **Kirstin Greaves-Lord**[5], **Joke H. M. Tulen**[6], **Karin Roelofs**[1,7], **Jan van der Ende**[5]

**1** Behavioural Science Institute, Radboud University, Nijmegen, The Netherlands, **2** Centre for Research on Child and Adolescent Mental Health, Karlstad University, Karlstad, Sweden, **3** Section of Clinical Developmental Psychology, Amsterdam Public Health Research Institute, Vrije Universiteit Amsterdam, Amsterdam, The Netherlands, **4** School of Health and Education, University of Skövde, Skövde, Sweden, **5** Department of Child and Adolescent Psychiatry/Psychology, Erasmus University Medical Center, Rotterdam, The Netherlands, **6** Department of Psychiatry, Erasmus University Medical Center, Rotterdam, The Netherlands, **7** Donders Institute for Brain Cognition and Behaviour, Radboud University, Nijmegen, The Netherlands

* brittany.evans@kau.se

**Data Availability Statement:** The data that support the JOiN study are available from the Erasmus University Medical Center but restrictions apply to the availability of these data, which were used

## Abstract

Growing up in an urban area has been associated with an increased chance of mental health problems in adults, but less is known about this association in adolescents. We examined whether current urbanicity was associated with mental health problems directly and indirectly via biological stress system functioning. Participants ($n$ = 323) were adolescents from the Dutch general population. Measures included home and laboratory assessments of autonomic nervous system and hypothalamic-pituitary-adrenal axis functioning, neighborhood-level urbanicity and socioeconomic status, and mother- and adolescent self-reported mental health problems. Structural equation models showed that urbanicity was not associated with mental health problems directly. Urbanicity was associated with acute autonomic nervous system and hypothalamic-pituitary-adrenal axis reactivity such that adolescents who lived in more urban areas showed blunted biological stress reactivity. Furthermore, there was some evidence for an indirect effect of urbanicity on mother-reported behavioral problems via acute autonomic nervous system reactivity. Urbanicity was not associated with overall autonomic nervous system and hypothalamic-pituitary-adrenal axis reactivity or basal hypothalamic-pituitary-adrenal axis functioning. Although we observed some evidence for associations between urbanicity, biological stress reactivity and mental health problems, most of the tested associations were not statistically significant. Measures of long-term biological stress system functioning may be more relevant to the study of broader environmental factors such as urbanicity.

## Introduction

Adolescence marks a unique period in human development when individuals transition from parent-protected childhood to social independence. This transition is facilitated by major psychosocial, hormonal, and neuronal maturational changes [1–4], which allow adolescents to

under license for the current study, and so are not publicly available. Data are however available from the authors upon reasonable request and with permission of the Erasmus University Medical Center. This restriction has been imposed by the Erasmus University Medical Ethics Committee. Data can be requested from the final author, Jan van der Ende (jan.vanderende@erasmusmc.nl), who will facilitate the data request, and/or the Erasmus University Medical Center Ethics Committee directly (metc@erasmusmc.nl).

**Funding:** The study was supported by The Netherlands Organisation for Health Research and Development (ZonMW; https://www.zonmw.nl) grant 3116.0002 awarded to ACH and by a Behavioural Science Institute Fellowship (Radboud University; https://www.ru.nl/bsi/) granted to BEE. The funders had no role in study design, data collection and analysis, decision to publish, or preparation of the manuscript.

**Competing interests:** The authors have declared that no competing interests exist.

gain the experiences and competencies they need to navigate the adult world. However, this plasticity also taxes adolescents with a heightened vulnerability to mental health problems [5, 6]. This normative growth and heightened vulnerability are driven, in part, by an increase in exploratory and risk-taking behavior [7]. Due to this increased exploratory behavior, broader socio-environmental influences such as the neighborhood may exert stronger effects on adolescents relative to children [8, 9].

Given adolescents' sensitivity to the broader social environment [8], neighborhood effects such as socioeconomic conditions, urbanicity, exposure to violence, social norms and institutional resources may be especially relevant for their development [10]. Living in an urban compared to a rural area, notably, seems to be associated with an increased risk of mental health problems [11]. This effect seems to be independent of other known risk factors such as sex, ethnicity, socioeconomic status and drug use [12]. Evidence exists for both a causal effect of urbanicity on mental health [12] as well as for the selective migration of individuals at risk for mental health problems toward more urban areas, which seems to be in part genetically driven [13]. Thus, the urbanicity-mental health association seems to be a combination of reciprocal influences between individuals and the wider social environment [14].

## Urbanicity and mental health

There is a substantial body of literature showing that adult inhabitants of urban areas are more likely to be diagnosed with a psychiatric disorder compared to those living in more rural areas [11, 15–17]. Retrospective studies showed that this increased risk for mental health problems may be even greater for those who grew up in a city [18], suggesting that the effects of an urban environment on mental health may be particularly influential during youth. A few studies in children corroborated this, demonstrating that those living in urban areas were more likely to be diagnosed with autism spectrum disorder and attention deficit disorder [19–22], to exhibit sub-clinical behavioral and emotional problems [23, 24], and to report symptoms of psychosis [25] compared to those living in more rural areas. Importantly, these effects remained when controlling for other individual-, family- and neighborhood-related confounders such as socioeconomic status, parental psychopathology and neighborhood cohesion [23, 25]. The previous research suggests that the association is not specific to certain psychiatric disorders, rather, that it pertains to a broad range of problems, both behavioral and emotional in nature.

A number of studies showed that urbanicity might affect adolescents' mental health as well. One of the first studies on neighborhood effects on youth showed that rates of juvenile delinquency were higher in urban areas [26]. Since then, researchers reported that adolescents living in urban areas were more likely to be diagnosed with a psychiatric disorder [27], and had symptoms of depression (especially in females) [28], aggression [29], and psychosis (if they had pre-existing psychotic symptoms) [30] compared to those living in more rural areas. However, others reported no association between urbanicity and depression in adolescents [31, 32]. The consensus from studies in children and adults seems to be that those living in urban areas are more likely to have (sub-clinical symptoms of) mental health disorders than those living in areas that are more rural. Regarding adolescents, the evidence is more mixed.

## Social stress in urban areas

Despite the evidence of a higher incidence of mental health problems in urban areas, the mechanisms underlying this association are not well understood. One of the most prominent hypotheses at this point is that social stress is greater in cities [33]. Social stress in humans is elicited by, for example, a crowded environment [34], greater anonymity [35], competition for

resources [36], perceived isolation [37], encounters with strangers and unclear dominance order [38], all of which may be more common in urban areas. Due to these factors, the threat of social evaluation and defeat increases [36, 39]. These social stress factors, in turn, are strongly predictive of a greater risk for mental health problems [40].

## The biological stress systems

Humans process social stress by activation of the biological stress systems. Moreover, biological stress system functioning has been put forth as a potential mechanism underlying the association between social-environmental factors, such as urbanicity, and mental health [14]. Humans are equipped with two major biological stress systems: the hypothalamic-pituitary-adrenal (HPA) axis and the autonomic nervous system (ANS). The production of cortisol, the end-product of the HPA axis, follows a circadian rhythm, with a peak approximately 30 minutes after awakening (i.e. the cortisol awakening response) followed by a gradual decline across the day [41]. The HPA axis and ANS are activated when an individual encounters a stressor, allowing the individual to respond adaptively. Catecholaminergic activation of the ANS is quick and serves the 'fight or flight' response and can be detected by, for example, an increase in heart rate [42]. The HPA axis responds more slowly and can be detected by increased levels of cortisol approximately 20 minutes after onset of the stressor [41].

Biological stress reactivity can be measured in different ways. Of these, acute and overall stress reactivity are widely used. Acute stress reactivity is an indication of the strength of the biological stress response. It may be characterized by more inter-individual variation as it is calculated by subtracting the lowest resting cortisol or heart rate value (during rest) from the highest cortisol or heart rate value (during stress) [43]. Overall stress reactivity is a measure of the overall pattern of responsivity during stress and is indexed by calculating the area under the curve (AUC) [44]. All biological data are included in the measure (from rest and stress periods), therefore some of the biological recovery from stress, i.e., the decline in stress levels following a stressor, may be included in this measure.

Although most individuals will respond biologically to a stressor, there are substantial between-person differences in this responsivity. Hyper-responsivity (exaggerated responding to a stressor and/or flatter cortisol curves across the day) and hypo-responsivity (blunted responding to a stressor) are both considered to indicate stress system dysregulation [45], and have been associated with mental health problems [46, 47]. During adolescence, the stress systems undergo developmental changes such that adolescents show heightened HPA axis reactivity and quicker ANS recovery than children and adults [48–52].

## Urbanicity and biological stress system functioning

Living in a stressful environment is known to disrupt the normal functioning and development of the stress systems [40, 53]. Urban environments may be more stressful than rural environments, due to heightened social stress in these places [33]. There is some evidence that living in an urban environment is associated with dysregulated stress system functioning. Results from neuroimaging studies showed that adults who currently lived in and grew up in a city showed differential limbic brain area responsivity to psychosocial stress compared to adults who lived in and had grown up in rural areas and small towns [54–56]. Furthermore, urban upbringing in adults was associated with a blunted cortisol awakening response and dysregulated HPA axis responses compared to growing up in rural areas and small towns [57]. Similarly, adults living in urban areas showed dysregulated ANS responses to a speech stressor compared to adults living in rural areas [58].

Very little is known about the association between urbanicity and stress system functioning in youth. In a study on determinants of biological stress system reactivity in youth, in the same sample as was used in the current study, we observed a fairly strong association between living in more urban areas and blunted HPA axis and heart rate reactivity in adolescents [59]. A recent study in a different sample showed no association between urbanicity and HPA axis functioning in children [60].

## The current study

In sum, previous research showed that living in a more urban environment may be associated with an increased risk for mental health problems (arrow C, Fig 1). Moreover, living in an urban area may contribute to dysregulation of the biological stress systems (arrow A, Fig 1). In light of evidence that biological stress system dysregulation is related to risk for mental health problems in youth, in particular to behavioral problems (arrow B, Fig 1) [61, 62], we proposed that dysregulated biological stress system functioning underlies the association between urbanicity and mental health problems (A & B > C). This hypothesis was recently examined in children, and rejected [60, 63]. However, these associations have not yet been examined in adolescents. Many developmental changes take place during adolescence. Specifically, adolescents show increased exploratory behavior and susceptibility to influence from the broader social environment [8], therefore urbanicity may be particularly relevant for their mental health. Biological stress system maturation [51], and general neurophysiological plasticity [6] during adolescence contribute to heightened vulnerability to mental health problems. Furthermore, evidence suggests that puberty (which occurs during or just prior to adolescence) may be a sensitive period during which the biological stress system becomes attuned to the environment [64]. Research on the association between urbanicity and mental health also showed that urbanicity had a stronger effect in youth than in adults [65]. Thus, it is possible that associations between urbanicity, stress system functioning and mental health are particularly relevant during adolescence.

In the current study, we aimed to examine these associations in a sample of adolescents from the Dutch general population. Using cross-sectional data, we utilized a path analysis to

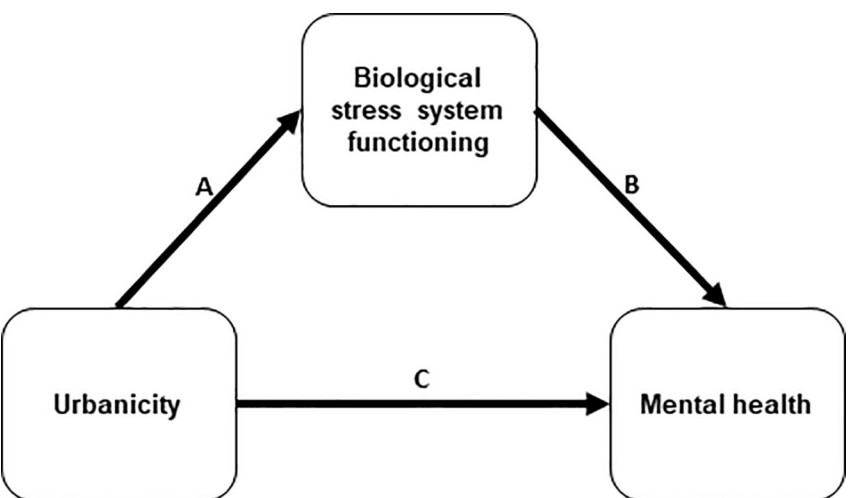

**Fig 1. Proposed model of the current study.** The proposed model was tested using two indices of mental health (behavioral and emotional problems, each reported by adolescents themselves and their mothers) and five indices of biological stress system functioning (overall ANS reactivity, acute ANS reactivity, overall HPA axis reactivity, acute HPA axis reactivity and basal HPA axis functioning). Urbanicity was measured at the neighborhood level, based on the surrounding address density.

examine whether urbanicity was associated with sub-clinical mental health problems (i.e. behavioral and emotional problems) directly, and indirectly via biological stress system functioning (i.e. basal HPA axis functioning, and HPA axis and ANS reactivity). We hypothesized that living in a more urban area would be directly associated with exhibiting more mental health problems directly, and indirectly via biological stress system functioning. We controlled for neighborhood- and family-level socioeconomic status, and thus tested proposed effects of urbanicity on biological stress system functioning and mental health independent of socioeconomic conditions. The research questions, methodology and analyses for the current study were peer-reviewed and pre-registered at the Open Science Framework [66].

## Materials and methods

### Participants

Adolescents were part of a larger longitudinal general population study in the Netherlands that examined the development of behavioral and emotional problems in youth (JOiN study) [67, 68]. Participants were randomly drawn from the municipal registers of 35 municipalities in the province of South Holland, the Netherlands (see S1 Fig for a flow chart). In this study, we used data from the first (T1, $N = 1710$) and second (T2, $N = 990$) assessment waves. We used primarily data from T2 in the current study, with the exception of family socioeconomic status which was assessed at T1 and address information at T1 which was used to exclude participants who had moved between T1 and T2. Complete data were available for 323 adolescents (see *Available data*). About half (54%) of the sample was female, the majority (86%) were of Dutch or other western background, and most (64%) were from a higher socioeconomic background.

### Procedure

The Erasmus University Medical Center Ethics Committee approved the study. Parents and adolescents gave informed consent and assent, respectively, at each measurement wave. T2 (November 2004-March 2009) took place between one and four years after T1 (December 2003-April 2005). At T2, adolescents and their parents completed questionnaires and adolescents collected saliva samples at home on a normal day (in order to measure basal HPA axis functioning) and participated in a psychosocial stress procedure (in order to measure HPA axis and ANS reactivity). The psychosocial stress procedure took place at the Erasmus University Medical Center in Rotterdam or at temporary testing locations nearer to adolescents' homes. It commenced with an explanation of the procedure by the experiment leader. After completing some questionnaires, the electrodes of the electrocardiogram were attached and participants were told to breathe normally and to relax. Electrocardio-activity was monitored constantly during the procedure. During the procedure, salivary cortisol was collected six times (see Fig 2). After a 10-minute pre-task rest period, the social stress tasks began, which were characterized by uncontrollability and social-evaluative threat and thus designed to elicit a biological stress response [69]. The tasks consisted of a mental arithmetic task (i.e. mental serial subtraction, four minutes), a public speaking task (eight minutes of mental preparation, six minutes of speech), and a computer mathematics task (numerical ordering, five minutes). Each task was performed in front of the test leader, and the speech was recorded on a digital camera. The session ended with a five-minute recovery period and a relaxing nature documentary (25 minutes). Perceived stress was self-reported five times during the procedure and used to determine whether the procedure was perceived as stressful.

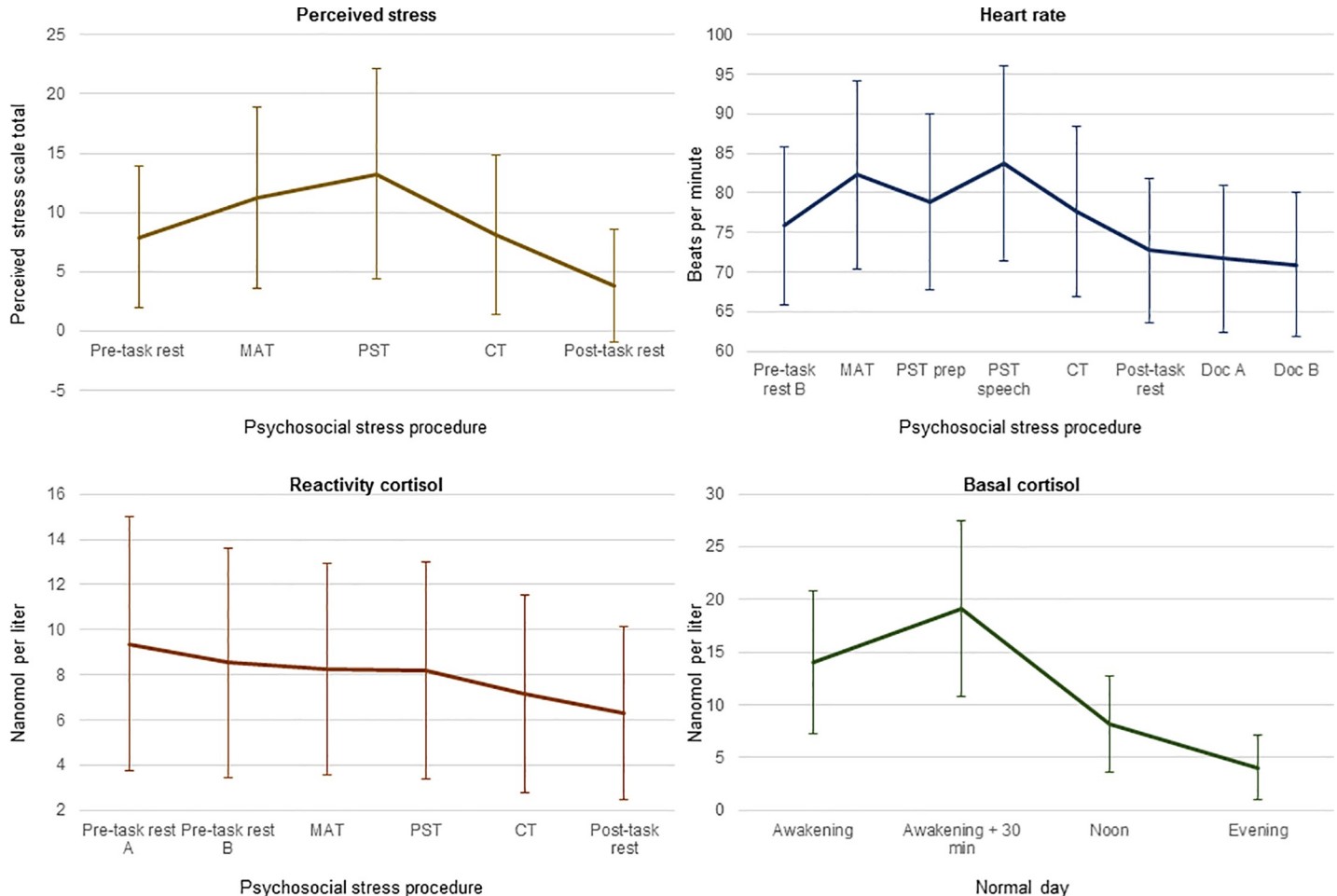

**Fig 2. Means and standard deviations of all perceived and (raw) biological stress indices.** Lines refer to means and error bars to standard deviations of all perceived (top left) and biological (heart rate: top right, reactivity cortisol: bottom left, basal cortisol: bottom right) stress indices. The evening measurement of basal cortisol was taken at 20.00. MAT = mental arithmetic task, PST = public speaking task, CT = computer task, prep = preparation, Doc = documentary.

## Measures

**Indices of mental health problems.** Mothers and adolescents completed the Dutch versions of the Child Behavior Checklist 6–18 [70] and the Youth Self Report [71], respectively. The versions contain 118 and 112 questions, respectively, that pertain to mental health problems and are answered on a scale of 0 to 2. The questionnaires were completed at home and brought to the university for the psychosocial stress procedure session. The index for *behavioral problems* consisted of the mean of the Attention Deficit Hyperactivity Problems, Oppositional Defiant Problems and Conduct Problems subscales. The index for *emotional problems* consisted of the mean of the Anxiety Problems, Depression Problems and Somatic Problems subscales. Tau-equivalent reliability coefficients (also known as Cronbach's alpha, as a measure of internal consistency) were .86 and .82 for behavioral problems and .84 and .83 for emotional problems on the mother- and self-reports, respectively.

**Current urbanicity.** Current urbanicity was measured at the neighborhood level. Neighborhoods are defined by Statistics Netherlands as part of one municipality and with a homogenous socioeconomic structure. In the Netherlands, neighborhoods consist of 1400 inhabitants on average (ranging between 35 and 28,380 as of 2018) [72]. Using the six-digit zip code of

participants' home address, we extracted data on the neighborhood they lived in for the year that they participated in the psychosocial stress procedure (1% of the sample participated in 2004, 1% in 2005, 17% in 2006 and 56% in 2007, 18% in 2008 and 7% in 2009) from Statistics Netherlands [73].

Current urbanicity was indicated by a five-category scale that was calculated by Statistics Netherlands using the surrounding address density (SAD). The SAD is a continuous measure indicating the degree of human activity in a given area [74]. It is based on the number of addresses within a one-kilometer radius around an address and is calculated by computing the mean SAD of all addresses in a neighborhood. Urbanicity is then coded on a scale from 0 (very rural; SAD < 500 addresses per $km^2$) to 4 (very urban; $\geq$ 2500 addresses per $km^2$). We utilized the categorical coding of urbanicity in order to improve the generalizability of our findings as this measure is commonly used [75–77] and recommended by Statistics Netherlands [74].

**Indices of biological stress system functioning.** To assess biological stress system functioning, we obtained five indices: two indices of ANS reactivity, two indices of HPA axis reactivity, and one index of basal HPA axis functioning. ANS reactivity was assessed by monitoring adolescents' heart rate (HR) constantly throughout the entire stress procedure. HR was measured using a three-lead electrocardiogram and a digital recorder (Vitaport[TM] System, TEMEC Instruments B.V., Kerkrade, the Netherlands). Data were imported and processed using a Vitascore[TM] software module (TEMEC Instruments B.V., Kerkrade, the Netherlands). This program calculated the interbeat intervals (IBI) of the electrocardiogram signal using R-top detection, resulting in IBI time series which were inspected for detection and removal of artifacts. HR time series were calculated from these IBI time series and expressed in beats per minute; the HR time series were subsequently averaged per period during the stress procedure. Fig 2 depicts when each stress measurement was taken during the stress procedure. An area under the curve with respect to increase (AUCiHR) [44] was calculated from the mean HR during each period of the stress procedure as an index of *overall ANS reactivity*. The AUCi was calculated if data were available for at least the pre-task rest period, one of the stressful tasks, and the recovery period. We calculated the maximum response for heart rate (MRHR) by subtracting the average heart rate during the pre-task rest from the maximum average heart rate during any of the three stressful tasks as an index of *acute ANS reactivity* [43, 78].

Salivary cortisol was collected six times during the psychosocial stress procedure by passive drooling. Taking into account the approximate 20-minute delay between activity in the hypothalamus and observable changes in salivary cortisol levels [41], the first two measurements (Reactivity Cortisol 1; RC1; and RC2) corresponded to cortisol levels during the pre-task period, RC3-RC5 corresponded to cortisol levels during each of the three stressful tasks, and RC6 corresponded to cortisol levels during the recovery period (see Fig 2). All sessions began in the early (approximately 12:00) or late (approximately 15:30) afternoon. All samples were sent collectively to a laboratory (Technical University Dresden, Dresden, Germany) for analysis. Cortisol levels greater than three standard deviations above the mean were removed due to potential contamination (e.g. by blood or medicine). An AUCi (AUCiC) was calculated as an index of *overall HPA axis reactivity* if the measurements RC1 or RC2, plus RC3 or RC4 or RC5, plus RC6 were available. The lower value of RC1 and RC2 was used in the calculation. The maximum response for cortisol (MRC) was calculated by subtracting the lower value of RC1 and RC2 from the highest value of RC3, RC4 or RC5 as an index of *acute HPA axis reactivity* [43, 79].

An additional four tubes were sent to participants' homes by mail. They were given detailed written and verbal instructions on the time and manner (passive drooling) of sample collections, and to preserve the tubes in the freezer until transportation to the university. Participants collected samples directly upon awakening (Basal Cortisol 1; BC1), 30 minutes thereafter (BC2), at 12:00 (BC3) and at 20:00 (BC4). From these samples, we calculated the area under

the curve with respect to ground (AUCgC) [44] in order to assess *basal HPA axis functioning*, as long as BC1, BC2 and BC4 were available. Our original pre-registered analysis plan included only the overall reactivity (AUCiC and AUCiHR) and basal HPA axis functioning (AUCgC) indices [66]. The acute stress reactivity measures were included post hoc, in light of newly published recommendations regarding stress reactivity indices [43].

**Covariates.** Neighborhood socioeconomic status, measured at the neighborhood level, was controlled for in all models. It was indicated by 15 characteristics of the neighborhood reported by Statistics Netherlands (e.g. average value of housing, proportion of persons with a high income). A principal components analysis of these variables was run in the larger JOiN sample population (all participants for whom address data was available at T2; *n* = 1105) in order to summarize them, resulting in two components that together explained 63% of the variance: employment (explaining 46% of the variance) and income (explaining 17% of the variance). The factor scores on these two components were used in the analysis.

We further controlled for family socioeconomic status, age and sex of the adolescent, and potential correlates of stress system functioning. Socioeconomic status of the family was based on the education level of both parents, reported by a parent at T1. We used the higher education level of either parent, categorized into three levels (i.e. 0 = completed primary education only or lower secondary education, 1 = completed higher secondary education, and 2 = completed college or university) based on the Dutch Education Level Division [80]. Sex and birth date (to calculate age at T2) of the adolescent were mother-reported.

We examined several potential covariates that could influence stress system functioning [59]. These included ethnicity of the adolescent (0 = Dutch or western immigrant background, 1 = non-western immigrant background; mother-reported), pubertal stage (using self-reported Tanner stages) [81], season of sampling (0 = spring/summer, 1 = autumn/winter; cortisol only), medicine and nicotine use (0 = no use, 1 = use; self- and mother-reported), whether girls had reached menarche and used oral contraceptives (0 = no; 1 = yes; self- and mother-reported), and girls' menstrual cycle phase (0 = follicular; 1 = luteal). Specifically for HPA axis and ANS reactivity, we examined body mass index (based on height and weight measured prior to the psychosocial stress procedure), whether adolescents had consumed caffeine, dairy products or engaged in physical exercise on the day of the psychosocial stress procedure (0 = no, 1 = yes; self- and mother-reported), and the time of day (0 = early, 1 = late afternoon) at which the psychosocial stress procedure took place. We determined which potential covariates would be included in the analyses pertaining to ANS and HPA axis functioning by calculating bivariate correlations between the stress indices and the potential covariates.

**Perceived stress.** Self-reported perceived stress was assessed five times during the psychosocial stress procedure (i.e. after the pre-task rest, each of the stressful tasks, and at the end of the procedure, see Fig 2) and used in a manipulation check to determine whether the procedure was perceived as stressful. Participants answered seven questions (see S1 File) using a visual thermometer (*Gevoelsthermometer* [Feelings thermometer]) ranging from 0 to 8 (available at https://www.pearsonclinical.nl/adis-c-complete-set). The scores were summed to a total score of perceived stress at each time point.

**Available data.** We included adolescents in the analyses if they lived in the same neighborhood between T1 and T2, in order to minimize within-subject variance in neighborhood-level variables. Of the 509 adolescents who participated in the T2 measurement, 34 moved to a different neighborhood and neighborhood information was not available from Statistics Netherlands for one individual, therefore these were excluded. At T2, outcome data (mother- and/or self-report) were available for 377 adolescents and 345 adolescents participated in the psychosocial stress procedure. After biological stress data cleaning, data were available on the predictor, outcome, and at least one of the stress variables for 323 adolescents, which made up our

final sample (see S1 Fig). Of the 509 adolescents who participated in T2, the adolescents included in the final sample differed from those who were not on a number of characteristics (i.e. more self-reported behavioral problems, higher neighborhood socioeconomic status and older age, see S1 Table).

## Statistical analysis

First, because some of the adolescents lived in the same neighborhood, we examined whether it was necessary to control for this similarity in environment. We calculated intraclass correlations (ICC) using empty models, in order to determine whether neighborhood explained sufficient variance in these measures (using the R [82] package 'Multilevel' [83]). As this was not the case (see S2 Table for details), we did not include neighborhood as a level in the analyses.

We then determined which potential covariates would be included in the analyses pertaining to ANS and HPA axis functioning by calculating bivariate correlations between the stress indices and all potential covariates (see section *Covariates* above). Variables that correlated significantly with the stress indices were included in the relevant analyses. We then ascertained whether the psychosocial stress procedure had induced perceived and biological stress by conducting a manipulation check by way of repeated-measures analysis of variance (ANOVA) on the perceived stress, heart rate and cortisol measurements across the psychosocial stress procedure.

Subsequently, we calculated descriptive statistics of and correlations between all variables, and checked for influential data points using Cook's distance (criterion $D < 1$). All variables were centered and standardized prior to the main analyses. The main analyses consisted of path analysis models in which the outcome measures of behavioral and emotional problems were estimated in separate models, and urbanicity was the main predictor. In each model, mother- and adolescent self-reported behavioral or emotional problems were estimated simultaneously. In each model (see Fig 1), we assessed whether urbanicity was associated with behavioral or emotional problems directly, and indirectly via biological stress system functioning (i.e. overall ANS reactivity, acute ANS reactivity, overall HPA axis reactivity, acute HPA axis reactivity and basal HPA axis functioning). In both models, we controlled for neighborhood socioeconomic status (employment and income), family socioeconomic status, sex and age of the adolescent, and any covariates related to stress system functioning. Because it has been suggested that socioeconomic conditions may underlie the association between urbanicity and mental health [14], we also ran the main analysis models without controlling for neighborhood and family socioeconomic status, as a supplementary analysis. To assess model fit, we used the Comparative Fit Index (CFI; with values > 0.95 indicating good fit), the Standardized Root Mean Squared Residual (SRMR; with values < 0.08 indicating good fit), and the Root Mean Square Error of Approximation (RMSEA; with values < 0.05 indicating good and < 0.01 indicating excellent fit [84]). Effects were considered significant at $p < .05$. We performed the main analyses in the lavaan package [85], version 0.6–5, of R [86] using full information maximum likelihood estimations. Subsequent to the main analyses, we performed a (post hoc) power analysis using the semPower package [87] of R. Given our RMSEA criteria of 0.05, alpha criteria of .05, sample size of 323, and degrees of freedom of 49 (model predicting behavioral problems) and 48 (model predicting emotional problems), we had a power of 0.93 and 0.92 for the models predicting behavioral and emotional problems, respectively.

## Results

### Manipulation check

Compared to the pre-task rest period, perceived stress and HR increased (see S3 Table). Cortisol levels did not increase, which was due to high cortisol levels during the pre-task rest, most

likely because of anticipation effects [52, 59, 88]. Cortisol levels did increase during the stress-ful tasks compared to the post-task resting period, which confirms a cortisol response.

## Preliminary results

Descriptive statistics of and correlations between all variables are portrayed in Table 1 and S4 Table, respectively. Descriptive statistics of all raw biological and perceived stress variables are illustrated in Fig 2. Percentages of participants in each of the five categories of the urbanicity scale are given in S2 Fig. In both models, we controlled for neighborhood-level socioeconomic status (employment and income), family socioeconomic status, sex and age. Biological stress system covariates were sex and pubertal stage for ANS reactivity; body mass index for acute ANS reactivity; sex, age, and test day exercise for overall HPA axis reactivity; age and season for acute HPA axis reactivity; and sex for basal HPA axis functioning.

## Model 1: Behavioral problems

In the first model, we tested the effects of urbanicity on self- and mother-reported behavioral problems directly, and indirectly via biological stress system functioning. The model had good fit (CFI = 0.95, SRMR = 0.04, RMSEA = 0.03). Urbanicity was not significantly associated with behavioral problems, neither directly (arrow C, Fig 1) nor indirectly via most of the biological stress variables (see Table 2). One indirect effect was statistically significant: urbanicity was indirectly related to mother-reported behavioral problems via acute ANS reactivity ($p$ = .04). The biological stress indices were not associated with behavioral problems (arrow B, Fig 1), with one exception: acute heart rate reactivity was associated with mother-reported behavioral problems such that adolescents who exhibited more behavioral problems showed weaker acute ANS reactivity. Urbanicity was significantly associated with two indices of stress system functioning such that adolescents who were from more urban areas showed weaker acute ANS and HPA axis reactivity (arrow A, Fig 1 and Tables 3 and 4). Urbanicity was not associated with overall ANS and HPA axis reactivity and basal HPA axis functioning. In a supplementary analysis in which we did not control for neighborhood and family socioeconomic status, the results were the same (see S5 Table, S6 Table and S7 Table).

## Model 2: Emotional problems

In the second model, we estimated the associations between urbanicity and self- and mother-reported emotional problems, directly and indirectly via biological stress system functioning. The model had good fit (CFI = 0.95. SRMR = 0.04. RMSEA = 0.03). Urbanicity was not associated with emotional problems, neither directly (arrow C, Fig 1) nor indirectly via biological stress system functioning (Table 5). Most of the stress indices were not associated with emotional problems, except overall HPA axis functioning. This association was such that adolescents whose mothers reported that they exhibited more emotional problems showed weaker overall HPA axis reactivity (arrow B, Fig 1). In a supplementary analysis in which we did not control for neighborhood and family socioeconomic status, the results were largely the same (see S8 Table).

## Discussion

In the current study, we examined whether urbanicity was associated with mental health problems directly and indirectly via biological stress system functioning in a sample of adolescents from the Dutch general population. We did not observe a direct association between urbanicity and mental health problems. There was some evidence for an indirect effect of urbanicity

**Table 1. Descriptive statistics for all variables.**

| | | Total sample | | | Boys | | | Girls | |
|---|---|---|---|---|---|---|---|---|---|
| | N | M (SD) or F (%) | Range | N | M(SD) or F (%) | Range | N | M(SD) or F (%) | Range |
| BP: self-report | 298 | 1.40(0.73) | 0.00–3.63 | 135 | 1.38(0.73) | 0.00–3.44 | 163 | 1.41(0.73) | 0.00–3.63 |
| EP: self-report | 298 | 0.89(0.68) | 0.00–3.72 | 135 | 0.62(0.56) | 0.00–3.49 | 163 | 1.11(0.69) | 0.00–3.72 |
| BP: mother-report | 304 | 0.78(0.71) | 0.00–3.44 | 139 | 0.76(0.64) | 0.00–2.66 | 165 | 0.81(0.75) | 0.00–3.44 |
| EP: mother-report | 304 | 0.57(0.58) | 0.00–2.85 | 139 | 0.42(0.46) | 0.00–2.85 | 165 | 0.71(0.63) | 0.00–2.70 |
| Urbanicity | 323 | 2.52(1.34) | 0.00–4.00 | 149 | 2.50(1.31) | 0.00–4.00 | 174 | 2.54(1.37) | 0.00–4.00 |
| Heart rate (AUCi) | 275 | 4639.88(609.83) | 3332.21–7209.23 | 120 | 4559.99(635.19) | 3332.21–6387.10 | 155 | 4701.74(584.03) | 3388.44–7209.23 |
| Heart rate (MR) | 277 | 10.21(8.59) | -10.69–39.95 | 121 | 9.38(8.17) | -10.69–31.94 | 156 | 10.85(8.87) | -4.16–39.95 |
| Cortisol (AUCi) | 301 | 329.23(212.61) | 44.69–1580.98 | 133 | 364.40(249.67) | 73.73–1580.98 | 168 | 301.39(173.69) | 44.69–1103.10 |
| Cortisol (MR) | 300 | 1.08(3.39) | -9.44–13.94 | 132 | 1.46(3.68) | -8.07–10.93 | 168 | 0.79(3.12) | -9.44–13.94 |
| Cortisol (AUCg) | 267 | 7154.37(2524.03) | 616.35–15428.40 | 128 | 6574.87(2507.84) | 616.35–15428.40 | 139 | 7688.02(2427.66) | 2557.20–14043.90 |
| SES Employment | 323 | 0.17(0.86) | -3.40–2.07 | 149 | 0.24(0.85) | -3.04–2.07 | 174 | 0.12(0.88) | -3.40–1.76 |
| SES Income | 323 | 0.05(0.95) | -2.17–3.69 | 149 | 0.13(0.96) | -1.85–3.69 | 174 | -0.02(0.93) | -2.17–3.31 |
| Family SES (l/a/h) | 323 | 11/25/64 | | 143 | 11/29/60 | | 164 | 11/22/67 | |
| Age | 323 | 17.07(1.51) | 13.00–20.83 | 149 | 17.15(1.36) | 13.00–20.83 | 174 | 17.01(1.62) | 13.00–20.67 |
| Sex (boy/girl) | 323 | 46/54 | | | | | | | |
| Pubertal stage | 289 | 4.34(0.68) | 2.00–5.00 | 137 | 4.26(0.67) | 2.00–5.00 | 152 | 4.42(0.68) | 2.00–5.00 |
| Body mass index | 314 | 21.73(3.27) | 16.00–40.56 | 142 | 21.24(2.93) | 16.00–31.44 | 172 | 22.14(3.48) | 16.65–40.56 |
| TD exercise (y/n) | 323 | 16/84 | | 149 | 85/15 | | 174 | 84/16 | |
| Season (sum/win) | 323 | 53/47 | | 149 | 52/48 | | 174 | 55/45 | |

F = frequency; BP = behavioral problems; EP = emotional problems; AUCi = area under the curve with respect to increase; MR = maximum response; AUCg = area under the curve with respect to ground; SES = socioeconomic status; l/a/h = low/average/high; TD = test day; y/n = yes/no; sum/win = spring and summer/autumn and winter; urbanicity, SES employment and SES income were measured at the neighborhood level.

on mother-reported behavioral problems via acute ANS reactivity (i.e. maximum heart rate response). The indirect effects via all other tested indices of biological stress system functioning were not statistically significant.

Recently, we examined the same research question as in the current study in a sample of children. In that study, we found no evidence for the hypothesized associations between current urbanicity and urban/rural upbringing. HPA axis functioning and mental health problems in two independent samples of children aged six to 12 years [60]. We then hypothesized that these associations may be particularly relevant during adolescence given the developmental changes that take place during this period. Specifically, adolescents may be more susceptible to influence from the broader social environment [8], the biological stress systems undergo maturational changes during adolescence [51], and increased neurophysiological plasticity contributes to a heightened vulnerability to mental health problems in adolescents [6]. In the current study, we did observe some evidence for associations between urbanicity, biological stress system functioning and behavioral problems, although the specific associations were not consistent across stress systems or indices of stress system functioning. Our findings thus provide some support for the idea that associations between urbanicity, stress system functioning and behavioral problems may be more salient during adolescence than in childhood.

Previous research in adults [11], children [25] and adolescents [28, 29] signaled a potential association between urbanicity and mental health. In the current study, we did not replicate this finding in adolescents. This is in line with a few other studies among adolescents [31, 32]. Furthermore, we did not observe this association in the children of the JOiN sample [60]. One reason for the lack of association in our study may be due to our use of a general population

**Table 2. Unstandardized estimates for structural equation models predicting behavioral problems.**

| | Self-report | | | | | Mother-report | | | | |
|---|---|---|---|---|---|---|---|---|---|---|
| | Est | SE | z | p | CI | Est | SE | z | p | CI |
| **Intercept** | 0.07 | 0.27 | 0.27 | .785 | -0.45/0.60 | -0.13 | 0.26 | -0.51 | .607 | -0.63/0.37 |
| **Indirect effects** | | | | | | | | | | |
| AUCiHR | 0.01 | 0.01 | 1.19 | .236 | -0.01/0.03 | 0.00 | 0.01 | 0.29 | .770 | -0.01/0.02 |
| MRHR | -0.00 | 0.01 | -0.04 | .972 | -0.02/0.02 | *0.03* | *0.02* | *2.03* | *.042* | *0.00/0.07* |
| AUCiC | 0.00 | 0.01 | 0.24 | .807 | -0.01/0.01 | 0.00 | 0.01 | 0.24 | .808 | -0.01/0.01 |
| MRC | 0.01 | 0.01 | 0.48 | .634 | -0.01/0.02 | 0.00 | 0.01 | 0.23 | .817 | -0.02/0.02 |
| AUCg | -0.00 | 0.00 | -0.33 | .738 | -0.01/0.01 | -0.00 | 0.00 | -0.34 | .731 | -0.01/0.01 |
| **Direct effects** | | | | | | | | | | |
| Urbanicity | -0.11 | 0.07 | -1.60 | .110 | -0.25/0.03 | -0.02 | 0.07 | -0.35 | .727 | -0.16/0.11 |
| AUCiHR | -0.13 | 0.07 | -1.83 | .068 | -0.27/0.01 | -0.02 | 0.07 | -0.30 | .767 | -0.16/0.12 |
| MRHR | 0.00 | 0.07 | 0.04 | .972 | -0.13/0.14 | **-0.21** | **0.07** | **-3.11** | **.002** | **-0.34/-0.08** |
| AUCiC | -0.12 | 0.07 | -1.58 | .115 | -0.26/0.03 | -0.09 | 0.07 | -1.32 | .188 | -0.24/0.05 |
| MRC | -0.03 | 0.07 | -0.48 | .629 | -0.16/0.10 | -0.02 | 0.07 | -0.23 | .817 | -0.14/0.11 |
| AUCg | 0.05 | 0.07 | 0.74 | .457 | -0.08/0.18 | 0.06 | 0.06 | 0.94 | .347 | -0.07/0.19 |
| SESem | -0.05 | 0.07 | -0.67 | .500 | -0.20/0.10 | -0.12 | 0.07 | -1.70 | .090 | -0.26/0.02 |
| SESin | -0.02 | 0.06 | -0.41 | .683 | -0.14/0.09 | 0.02 | 0.06 | 0.39 | .699 | -0.09/0.13 |
| SESfam | -0.05 | 0.12 | -0.37 | .711 | -0.28/0.19 | 0.05 | 0.12 | 0.44 | .658 | 0.18/0.29 |
| Sex | -0.02 | 0.12 | -0.14 | .889 | -0.26/0.22 | 0.05 | 0.12 | 0.44 | .663 | -0.18/0.28 |
| Age | -0.00 | 0.07 | -0.06 | .951 | -0.13/0.12 | -0.03 | 0.06 | -0.54 | .589 | -0.14/0.08 |

**Bold** indicates *p* < .01; *italics* indicates *p* < .05; AUCiHR = area under the curve with respect to ground, calculated for heart rate; MRHR = maximum heart rate response; AUCiC = area under the curve with respect to increase, calculated for cortisol; MRC = maximum cortisol response; AUCgC = area under the curve with respect to ground, calculated for cortisol; SESem = socioeconomic status, employment component; SESin = socioeconomic status, income component; SESfam = family socioeconomic status.

sample, in which we examined symptoms of mental health problems and not mental disorders. Most of the previously reported associations between urbanicity and mental health examined diagnoses of mental disorders as the outcome measure [11, 27, 30]. In our study, we were specifically interested in sub-clinical levels of mental health problems as less is known about this in relation to urbanicity. Our findings suggest that urbanicity is not related to these sub-clinical mental health problems in adolescents.

It is possible that the lack of association between urbanicity and mental health problems in adolescents also has to do with the majority of the adolescents in our sample coming from

**Table 3. Estimates for structural equation model pathways predicting autonomic nervous system functioning variables.**

| | Heart rate (AUCi) | | | | | Heart rate (MR) | | | | |
|---|---|---|---|---|---|---|---|---|---|---|
| | Est | SE | z | p | CI | Est | SE | z | p | CI |
| **Intercept** | -0.75 | 0.39 | -1.94 | .052 | -1.51/0.01 | 0.01 | 0.06 | 0.12 | .907 | -0.11/0.12 |
| **Direct effects** | | | | | | | | | | |
| Urbanicity | -0.09 | 0.06 | -1.58 | .113 | -0.21/0.02 | **-0.16** | **0.06** | **-2.69** | **.007** | **-0.27/-0.04** |
| Sex | 0.10 | 0.11 | 0.87 | .385 | -0.12/0.31 | - | - | - | - | - |
| Puberty | 0.14 | 0.08 | 1.68 | .092 | -0.02/0.31 | - | - | - | - | - |
| BMI | - | - | - | - | - | *-0.12* | *0.06* | *-2.12* | *.034* | *-0.23/-0.01* |

**Bold** indicates *p* < .01; *italics* indicates *p* < .05; AUCi = area under the curve with respect to increase; MR = maximum response; Puberty = pubertal stage; BMI = body mass index; urbanicity was measured at the neighborhood level. Estimates are as reported in the model predicting behavioral problems.

**Table 4. Estimates for structural equation model pathways predicting HPA axis functioning variables.**

| | Cortisol (AUCi) | | | | | Cortisol (MR) | | | | | Cortisol (AUCg) | | | | |
|---|---|---|---|---|---|---|---|---|---|---|---|---|---|---|---|
| | Est | SE | z | p | CI | Est | SE | z | p | CI | Est | SE | z | p | CI |
| **Intercept** | **0.79** | **0.22** | **3.60** | **.000** | **0.36/1.23** | *0.38* | *0.17* | *2.20* | *.028* | *0.04/0.71* | **-0.70** | **0.19** | **-3.70** | **.000** | **-1.07/-0.33** |
| **Direct effects** | | | | | | | | | | | | | | | |
| Urbanicity | -0.01 | 0.06 | -0.25 | .805 | -0.12/0.10 | *-0.14* | *0.06* | *-2.44* | *.015* | *-0.26/-0.03* | -0.02 | 0.06 | -0.37 | .710 | -0.14/0.09 |
| Age | 0.03 | 0.05 | 0.60 | .551 | -0.07/0.13 | -0.09 | 0.06 | -1.40 | .162 | -0.21/0.03 | - | - | - | - | - |
| Sex | *-0.23* | *0.10* | *-2.30* | *.022* | *-0.42/-0.03* | - | - | - | - | - | **0.46** | **0.12** | **3.91** | **.000** | **0.23/0.70** |
| TD exercise | **-0.38** | **0.13** | **-3.00** | **.003** | **-0.63/-0.13** | - | - | - | - | - | - | - | - | - | - |
| Season | - | - | - | - | - | *-0.25* | *0.11* | *-2.25* | *.024* | *-0.46/-0.03* | - | - | - | - | - |

**Bold** indicates $p < .01$; *italics* indicates $p < .05$; AUCi = area under the curve with respect to increase; MR = maximum response; AUCg = area under the curve with respect to ground; TD exercise = test day exercise; urbanicity was measured at the neighborhood level. Estimates are as reported in the model predicting behavioral problems.

families with relatively high socioeconomic status. Only 11% of the adolescents were from low socioeconomic status families, as indicated by parent education levels. Higher family socioeconomic status may act as a protective factor for mental health problems through, for example, higher quality housing (with e.g. less crowding inside the house, better insolation against noise) [89, 90], residency in neighborhoods with more green areas [91], or parenting. There is

**Table 5. Unstandardized estimates for structural equation models predicting emotional problems.**

| | Self-report | | | | | Mother-report | | | | |
|---|---|---|---|---|---|---|---|---|---|---|
| | Est | SE | z | p | CI | Est | SE | z | p | CI |
| **Intercept** | **-1.31** | **0.25** | **-5.16** | **.000** | **-1.81/-0.81** | **-0.89** | **0.25** | **-3.53** | **.000** | **-1.38/-0.40** |
| **Indirect effects** | | | | | | | | | | |
| AUCiHR | -0.00 | 0.01 | -0.51 | .609 | -0.02/0.01 | -0.00 | 0.01 | -0.28 | .779 | -0.02/0.01 |
| MRHR | 0.00 | 0.01 | 0.02 | .987 | -0.02/0.02 | 0.02 | 0.01 | 1.44 | .151 | -0.01/0.04 |
| AUCiC | 0.00 | 0.01 | 0.30 | .766 | -0.01/0.01 | 0.00 | 0.01 | 0.30 | .763 | -0.02/0.02 |
| MRC | -0.01 | 0.01 | -0.80 | .427 | -0.03/0.01 | -0.01 | 0.01 | -0.97 | .332 | -0.03/0.01 |
| AUCg | -0.00 | 0.00 | -0.20 | .845 | -0.00/0.00 | -0.00 | 0.00 | -0.36 | .717 | -0.01/0.01 |
| **Direct effects** | | | | | | | | | | |
| Urbanicity | -0.07 | 0.07 | -1.07 | .286 | -0.20/0.06 | -0.09 | 0.07 | -1.30 | .195 | -0.21/0.04 |
| AUCiHR | 0.04 | 0.07 | 0.54 | .589 | -0.10/0.17 | 0.02 | 0.07 | 0.29 | .775 | -0.12/0.16 |
| MRHR | -0.00 | 0.07 | -0.02 | .987 | -0.13/0.13 | -0.11 | 0.07 | -1.70 | .089 | -0.24/0.02 |
| AUCiC | -0.10 | 0.07 | -1.46 | .146 | -0.24/0.04 | *-0.15* | *0.07* | *-2.23* | *.026* | *-0.29/-0.02* |
| MRC | 0.05 | 0.06 | 0.84 | .401 | -0.07/0.18 | 0.07 | 0.06 | 1.06 | .291 | -0.06/0.19 |
| AUCg | 0.01 | 0.06 | 0.22 | .826 | -0.11/0.14 | 0.07 | 0.07 | 1.09 | .275 | -0.06/0.20 |
| SESem | -0.07 | 0.07 | -1.02 | .307 | -0.21/0.07 | **-0.19** | **0.07** | **-2.75** | **.006** | **-0.32/-0.05** |
| SESin | -0.08 | 0.06 | -1.36 | .173 | -0.18/0.03 | 0.03 | 0.06 | 0.55 | .583 | -0.08/0.14 |
| SESfam | 0.16 | 0.12 | 1.36 | .175 | -0.07/0.39 | 0.15 | 0.12 | 1.27 | .204 | -0.08/0.38 |
| Sex | **0.69** | **0.12** | **6.02** | **.000** | **0.47/0.92** | **0.45** | **0.12** | **3.94** | **.000** | **0.23/0.68** |
| Age | 0.08 | 0.06 | 1.36 | .173 | -0.04/0.20 | 0.09 | 0.06 | 1.52 | .127 | -0.02/0.19 |

**Bold** indicates $p < .01$; *italics* indicates $p < .05$; AUCiHR = area under the curve with respect to ground. calculated for heart rate; MRHR = maximum heart rate response; AUCiC = area under the curve with respect to increase. calculated for cortisol; MRC = maximum cortisol response; AUCgC = area under the curve with respect to ground. calculated for cortisol; SESem = socioeconomic status. employment component; SESin = socioeconomic status. income component; SESfam = family socioeconomic status.

a robust association between parental education and parenting, such that parents with higher educational levels tend to exhibit higher quality caregiving [92–94]. Thus, the majority of adolescents in our sample may have been buffered by advantages at several levels that come with having a higher socioeconomic status background.

In our study, we examined five indices of biological stress system functioning in order to examine different aspects of the stress system in adolescents in relation to urbanicity and mental health. Perhaps not surprising given the wealth of mixed findings in the biological stress literature, the associations tested in our study were not consistent across the different indices of stress system functioning. Basal HPA-axis functioning is an index of how well the HPA axis functions on a daily basis, which is important for physical and mental health [95]. In our study, basal HPA axis functioning was not associated with urbanicity or mental health. Our measures of biological stress reactivity, on the other hand, were somewhat associated with both urbanicity and mental health.

Measures of biological stress reactivity indicate how well the stress systems respond to challenge [96] and can be calculated in a number of ways. Of these, acute (i.e. maximum stress response) and overall (i.e. AUCi calculation) stress reactivity are widely used. Acute stress reactivity is an indication of the strength of the biological stress response. The recommendation from recent work on the HPA axis response is to use this maximum stress response as an index of stress reactivity [43]. In our study, urbanicity was directly associated with acute ANS and HPA-axis reactivity, but not overall reactivity. Similarly, acute ANS reactivity was strongly associated with mother-reported behavioral problems. In terms of both ANS and HPA axis reactivity, acute reactivity (maximum stress response) seems to reflect the biological stress response most accurately [43, 78]. For example, acute HPA axis reactivity seems to be less influenced by HPA axis secretory activity that is not related to stress, compared to measures of overall HPA axis reactivity [43]. Therefore, our study seems to provide some evidence for associations between urbanicity and biological stress system functioning on the one hand, and between biological stress system functioning and behavioral problems on the other hand.

AUCi calculations are another index of reactivity to stress, yet are distinct from maximum stress response indicators as they provide a measure of the overall pattern of responsivity during stress. We found that HPA-axis reactivity was associated with mother-reported emotional problems, however, this association was weak. Overall stress reactivity was not associated with urbanicity in our study. That different indices of biological stress reactivity are distinct constructs is furthermore reflected, in our study, in the bivariate correlations between the biological stress reactivity indices, which were not particularly high ($r$ between .18 and .36). Thus, our findings highlight the relevance of including multiple indices of stress reactivity in order to work toward a comprehensive understanding of biological stress system functioning in relation to mental health and the broader social environment.

Related to our observation that urbanicity was differentially related to particular indices of biological stress system functioning, the two previous studies in adults that examined associations between urbanicity and biological stress system (re)activity also found different effects for different indices of biological stress. Armstead and colleagues [58] found that urban residents showed greater heart rate reactivity (using change scores between baseline and stressful tasks) compared to rural residents, whereas we found that urban adolescents showed blunted heart rate reactivity compared to rural adolescents (Table 3 and [59]). Steinheuser and colleagues [57] reported that urban upbringing compared to rural upbringing was associated with greater HPA axis reactivity to stress but a blunted cortisol awakening response. Thus, in the few studies that have examined associations between urbanicity and biological stress system functioning, the direction of effects within and across stress systems has not been consistent. This could be partly due to some studies assessing current urbanicity and others assessing

urban upbringing. The inconsistency is also similar to findings from the literature on the association between socioeconomic status and different measures of biological stress system functioning [97–99]. Possibly, measures of long-term biological stress system functioning (e.g. hair cortisol concentrations) are more relevant in the investigation of the effects of broader, long-term environmental factors such as urbanicity and socioeconomic status. Indeed, associations between socioeconomic status and long-term HPA axis functioning have been increasingly reported in the literature [100–102]. Thus, such measures of long-term stress system functioning may be more appropriate to the study of long-term environmental stressors.

In our study, most of the indices of biological stress system functioning were not associated with mental health problems in adolescents (arrow B, Fig 1), with two exceptions. Acute ANS reactivity was associated with mother-reported behavioral problems and overall HPA-axis reactivity was associated with mother-reported emotional problems. Thus, biological stress reactivity was more strongly related to behavioral problems than to emotional problems. This is consistent with previous research: blunted biological stress reactivity has been consistently associated with behavioral problems [61, 103]. Earlier findings pertaining to emotional problems have been more mixed [104–106]. The results from our study underline the association between biological stress reactivity, particularly ANS reactivity, and behavioral problems.

One of the critical issues in our study was to disentangle the effects of urbanicity and socioeconomic status by controlling for neighborhood- and family-level socioeconomic status. Urbanicity and socioeconomic status are closely related [107], and most previous studies have been unable to systematically examine the separate effects of each. In our study, while urbanicity was not directly associated with mental health, neighborhood socioeconomic status was, such that individuals from lower socioeconomic status neighborhoods exhibited more mother-reported emotional problems. This is consistent with a large body of literature on the effects of neighborhood socioeconomic status on youth [108, 109]. Interestingly, family socioeconomic status was not associated with mental health in youth in our study. The literature regarding family versus neighborhood socioeconomic status effects on youth mental health has been inconsistent, with some reports of neighborhood effects being more salient than family effects [110], and other reports of family socioeconomic status accounting for the effects of neighborhood socioeconomic status [111, 112]. Many of the inconsistencies may be attributable to differences in measurement and analyses (e.g. use of single- or multilevel modeling), and further research is needed to examine the associations, and potential interactions, between urbanicity, socioeconomic status and mental health at the individual/family and neighborhood levels. As a supplementary analysis, we ran our main analysis models without controlling for family and neighborhood socioeconomic status, as socioeconomic conditions have been suggested to underlie the association between urbanicity and mental health [14]. This supplementary analysis showed that the results remained largely the same as in the models controlling for socioeconomic status, confirming that, in our study, the effects of urbanicity seem to be independent of socioeconomic conditions.

The findings from this study should be taken in light of some important limitations. As mentioned above, the majority of our sample consisted of adolescents from higher socioeconomic status families, and thus the findings may not be generalizable to youth from lower socioeconomic status backgrounds. In addition, the age range was large, which hampered our ability to examine developmental effects. Furthermore, our data was cross-sectional in nature. We included only adolescents who had lived in the same neighborhood for at least the past year (since the T1 measurement), in this way strengthening the robustness of our main predictor, however we were not able to account for longer-term neighborhood effects. Also, we used 'administrative' neighborhoods as defined by Statistics Netherlands in order to obtain objective measures of the neighborhood. Neighborhoods are defined based on areas with a

homogenous socioeconomic structure [72], but they may differ from 'natural' neighborhoods, or neighborhoods as perceived by the inhabitants, which may be more ecologically valid. Also regarding the neighborhood, our measure of urbanicity was not evenly distributed across the five categories, as most individuals resided in urban or very urban neighborhoods. This uneven distribution may have affected our analyses. In addition, we did not account for the association between behavioral and emotional problems, as we examined these outcomes in separate models. Finally, we would like to emphasize that our study is just one part of a larger picture regarding the associations between the broader social environment, biological stress system functioning and mental health among adolescents. In our analyses, we did not include many complex associations and interactions that unequivocally affect these and related factors.

## Conclusions

Mental health problems have been posited to be more prevalent in individuals living in more urban areas. We examined biological stress system functioning as a potential underlying mechanism of this association in adolescents. There was some evidence for an indirect effect of urbanicity on mother-reported behavioral problems via blunted acute ANS reactivity. However, most of the indirect effects tested were not statistically significant. Possibly, the relatively high family socioeconomic status of most adolescents in our sample buffered the effects of urbanicity on biological stress system functioning and mental health. Alternative measures of biological stress such as long-term biological stress levels may be more suited to examining the effects of the broader environment on youth.

## Supporting information

**S1 Fig. Flow chart describing the available data. ANS = autonomic nervous system, HPA = hypothalamic-pituitary-adrenal.**
(TIF)

**S2 Fig. Percent of participants from each of the five categories of urbanicity as defined by Statistics Netherlands.** The urbanicity score is calculated using the surrounding address density (SAD) and coded as very rural (average SAD $< 500$ addresses per km$^2$), rural (average SAD between 500 and 1000 addresses per km$^2$), town (average SAD between 1000 and 15000 addresses per km$^2$), urban (average SAD between 1500 and 2500 addresses per km$^2$), and very urban (average SAD $\geq 2500$ addresses per km$^2$). In the analyses, the urbanicity scale was utilized as a continuous measure.
(TIF)

**S1 File. Perceived stress questionnaire.** Participants were shown a feelings thermometer (available at https://www.pearsonclinical.nl/adis-c-complete-set) as they were asked these questions. Translation was for the purpose of providing supplementary information and was not done systematically.
(DOCX)

**S1 Table. Differences between samples of those included in the final models and those who were not included in the final models.** AUCi = area under the curve with respect to increase; MR = maximum response; AUCg = area under the curve with respect to ground; SES = socioeconomic status; l/a/h = low/average/high. For continuous variables (i.e. behavioral problems, emotional problems, urbanicity, all biological stress variables, neighborhood SES variables and age) $t$ and $d$ statistics are reported, for categorical variables (i.e. family SES and sex) $\chi^2$ and $\varphi$ statistics are reported.
(DOCX)

**S2 Table. Intraclass correlations (1 and 2) for all outcome and intermediary variables.**
Intraclass correlations (ICC) were calculated using empty models. The ICC1 indicates the percentage of variance in behavioral and emotional problems that can be explained by group (i.e. neighborhood) membership. The ICC2 is an indication of reliability and should be > .70 [113]. AUCiHR = area under the curve with respect to ground, calculated for heart rate; MRHR = maximum heart rate response; AUCiC = area under the curve with respect to increase, calculated for cortisol; MRC = maximum cortisol response; AUCgC = area under the curve with respect to ground, calculated for cortisol.
(DOCX)

**S3 Table. Manipulation check statistics from repeated measures analyses of variance testing whether the psychosocial stress procedure induced perceived and biological stress.** All statistics are Greenhouse-Geisser corrected. Main effects of time: perceived stress model $df$ = 3.08, cortisol model $df$ = 2.43, heart rate model $df$ = 2.98. For all contrasts: $df$ = 1. Pre-task cortisol value pertains to the lower value of RC1 and RC2. MAT = mental arithmetic task; PST = public speaking task; prep = preparation; CT = computer task.
(DOCX)

**S4 Table. Zero-order correlations between all variables used in the main analyses. Bold** = p < .01; italics = p < .05. BP = behavioral problems; EP = emotional problems; self and mother indicate the informant; AUCiHR = area under the curve with respect to ground, calculated for heart rate; MRHR = maximum heart rate response; AUCiC = area under the curve with respect to increase, calculated for cortisol; MRC = maximum cortisol response; AUCgC = area under the curve with respect to ground, calculated for cortisol; SESem = socioeconomic status, employment component; SESin = socioeconomic status, income component; SESfam = family socioeconomic status; Puberty = pubertal stage; BMI = body mass index; TDexercise = test day exercise; urbanicity, SES employment and SES income were measured at the neighborhood level.
(DOCX)

**S5 Table. Unstandardized estimates for structural equation models predicting behavioral problems, not controlling for socioeconomic status. Bold** indicates $p < .01$; *italics* indicates $p < .05$; AUCiHR = area under the curve with respect to ground, calculated for heart rate; MRHR = maximum heart rate response; AUCiC = area under the curve with respect to increase, calculated for cortisol; MRC = maximum cortisol response; AUCgC = area under the curve with respect to ground, calculated for cortisol. Model fit indices were: CFI = .98, RMSEA = .03, SRMR = .04.
(DOCX)

**S6 Table. Estimates for structural equation model pathways predicting autonomic nervous system functioning variables, not controlling for socioeconomic status. Bold** indicates $p <$ .01; *italics* indicates $p < .05$; AUCi = area under the curve with respect to increase; MR = maximum response; Puberty = pubertal stage; BMI = body mass index; urbanicity was measured at the neighborhood level. Estimates are as reported in the model predicting behavioral problems, not controlling for socioeconomic status.
(DOCX)

**S7 Table. Estimates for structural equation model pathways predicting HPA axis functioning variables, not controlling for socioeconomic status. Bold** indicates $p < .01$; *italics* indicates $p < .05$; AUCi = area under the curve with respect to increase; MR = maximum

response; AUCg = area under the curve with respect to ground; TD exercise = test day exercise; urbanicity was measured at the neighborhood level. Estimates are as reported in the model predicting behavioral problems, not controlling for socioeconomic status.
(DOCX)

**S8 Table. Unstandardized estimates for structural equation models predicting emotional problems, not controlling for socioeconomic status. Bold** indicates $p < .01$; *italics* indicates $p < .05$; AUCiHR = area under the curve with respect to ground, calculated for heart rate; MRHR = maximum heart rate response; AUCiC = area under the curve with respect to increase, calculated for cortisol; MRC = maximum cortisol response; AUCgC = area under the curve with respect to ground, calculated for cortisol. Model fit indices were: CFI = .97, RMSEA = .03, SRMR = .04.
(DOCX)

## Acknowledgments

Data used in this study are from a large-scale longitudinal study (the JOiN study). conducted by the Erasmus University Medical Center. Data from the study have been used in research that has been published elswhere. We would like to thank all participants of the JOiN study and their parents. The JOiN study was conducted by the Department of Child and Adolescent Psychiatry/Psychology of the Erasmus University Medical Center.

## Author Contributions

**Conceptualization:** Brittany E. Evans, Anja C. Huizink, Karin Roelofs.

**Data curation:** Brittany E. Evans, Kirstin Greaves-Lord, Jan van der Ende.

**Formal analysis:** Brittany E. Evans, Jan van der Ende.

**Funding acquisition:** Brittany E. Evans, Anja C. Huizink.

**Investigation:** Brittany E. Evans, Kirstin Greaves-Lord.

**Methodology:** Joke H. M. Tulen.

**Project administration:** Brittany E. Evans, Anja C. Huizink, Jan van der Ende.

**Resources:** Anja C. Huizink.

**Visualization:** Brittany E. Evans.

**Writing – original draft:** Brittany E. Evans.

**Writing – review & editing:** Brittany E. Evans, Anja C. Huizink, Kirstin Greaves-Lord, Joke H. M. Tulen, Karin Roelofs, Jan van der Ende.

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
