## [Decision Letter · Decision Letter 0]

17 Sep 2019

PONE-D-19-18858

Urbanicity, biological stress system functioning and mental health in adolescents

PLOS ONE

Dear Dr. Evans,

Thank you for submitting your manuscript to PLOS ONE. After careful consideration, we feel that it has merit but does not fully meet PLOS ONE’s publication criteria as it currently stands. Therefore, we invite you to submit a revised version of the manuscript that addresses the points raised during the review process.

We would appreciate receiving your revised manuscript by Nov 01 2019 11:59PM. To enhance the reproducibility of your results, we recommend that if applicable you deposit your laboratory protocols in protocols.io, where a protocol can be assigned its own identifier (DOI) such that it can be cited independently in the future. For instructions see: http://journals.plos.org/plosone/s/submission-guidelines#loc-laboratory-protocols

We look forward to receiving your revised manuscript.

Kind regards,

Geilson Lima Santana, M.D., Ph.D.

Academic Editor

PLOS ONE

Journal Requirements:

1.

We note that you have indicated that data from this study are available upon request. PLOS only allows data to be available upon request if there are legal or ethical restrictions on sharing data publicly. For more information on unacceptable data access restrictions, please see http://journals.plos.org/plosone/s/data-availability#loc-unacceptable-data-access-restrictions.

Reviewers' comments:

Reviewer's Responses to Questions

**Comments to the Author**

1. Is the manuscript technically sound, and do the data support the conclusions?

Reviewer #1: Yes

Reviewer #2: Yes

Reviewer #3: Partly

2. Has the statistical analysis been performed appropriately and rigorously? 

Reviewer #1: Yes

Reviewer #2: Yes

Reviewer #3: No

3. Have the authors made all data underlying the findings in their manuscript fully available?

Reviewer #1: No

Reviewer #2: Yes

Reviewer #3: No

4. Is the manuscript presented in an intelligible fashion and written in standard English?

Reviewer #1: Yes

Reviewer #2: Yes

Reviewer #3: Yes

5. Review Comments to the Author

Reviewer #1: Summary

In this manuscript the authors investigate the association of urban upbringing and mental health is adolescents. An association previously observed in adults.

They opted for a structural equation modeling examining if urbanicity was associated with mental health problems directly and indirectly via biological stress system functioning (autonomic nervous system and hypothalamic-pituitary-adrenal axis) while controlling for the adequate covariates. The participants were dutch adolescents recruited from the general population.

The authors found urbanicity was not directly associated with mental health problems, but found associations for urbanicity acute autonomic nervous system and hypothalamic-pituitary-adrenal axis reactivity as well as other findings discussed in text.

The piece is technically sound, well written and the topic covered is thoroughly interesting. nevertheless some minor revisions are necessary for compliance with PLOS ONE editorial guidelines.

Major Issues

Page 26 - lines 605 - 607 - Acknowledgements: This session includes funding sources, in discordance with the submission guidelines: “Funding information should only be entered in the financial disclosure section of the submission system1.”

Page 32 - lines 759-761 - References: Reference #56 references unavailable or unpublished work, according to PLOS ONE submission guidelines this data should be included as supplementary material or deposited in a publicly available database1. The guideline includes manuscripts that have been submitted but not yet accepted on this category1.

Page 32 - lines 767-770 - References: References #59 and #60 are on the Open Science Framework, but are not readily available (available upon request only).

Page 12 - lines 312-313 - Measures: The content of content of the seven questions and the appearance of the visual thermometer is not revealed, which impairs the reproducibility of the experiment. Authors do inform methodology is pre-registered at the Open Science Framework, but it is not readily available.

Minor Issues

Page 2 - lines 30-31 - Abstract: It is not clear if what will be analyzed is the association in adolescents that grew up in urban areas or adolescents that are growing up in urban areas. It does become clear once we advance on the reading of the piece, but it would be interesting to clarify it from the start.

Page 3 - lines 71-73 - Introduction: The passage makes reference to mental health in general, but the paper referenced (reference #14) is specifically on schizophrenia. It would be interesting to add some more references for this claim and I am sure the authors have plenty of literature collected on that.

Page 8 - line 216 - Measures: A brief explanation on the importance of cronbach's alpha would be advisable as is may deter readers not familiar with this measure. It would be interesting to consider the use of the term tau-equivalent reliability as there is some discussion about the interchangeability of the terms2. Also, it may be interesting to consider a congeneric reliability analysis.

Page 9 - lines 229-231 - Measures: The passage states the categorical coding of surrounding address density (SAD) was chosen to improve generalizability and that this is common practice in literature, but reference a single paper (that is not a review). It would be interesting to add some references to substantiate this claim.

Page 9 - lines 232-234 - Measures: The passage introduces of biological stress system functioning utilized in the investigation (two indices of ANS reactivity: heart rate AUCi and MR; two indices of HPA axis reactivity: cortisol AUCi and MR; one index of basal HPA axis functioning: AUCgC). An early introduction of what these measures will be would improve the reading experience.

Page 12 - lines 324-325 - Measures: It may be interesting to consider alternatives p-values considering recent discussions on the subject 3,4. It also would be interesting to add some comments about effect size and/or measures of association for the analysis that were statistically significant.

Final comment

I believe the authors will have no problems addressing this minor changes and I make myself available in advance to look at a revised version of the article.

References

1. PLOS ONE: accelerating the publication of peer-reviewed science, https://journals.plos.org/plosone/s/submission-guidelines (accessed 1 August 2019).

2. Cho E. Making Reliability Reliable: A Systematic Approach to Reliability Coefficients. Organizational Research Methods 2016; 19: 651–682.

3. Wasserstein RL, Lazar NA. The ASA’s Statement on p-Values: Context, Process, and Purpose. Am Stat 2016; 70: 129–133.

4. Ioannidis JPA. The Proposal to Lower P Value Thresholds to .005. JAMA 2018; 319: 1429–1430.

Reviewer #2: This is an interesting study on the association between urbanicity and mental health, investigated in a direct way, and via biological stress system function. The association is known in adults, but in adolescents there is not much evidence yet. The authors have included many measures, to come to a more detailed view on this association, in which they succeed.

However, I have some questions and suggestions that need further attention. Especially the rationale for re-using a hypothesis that was rejected in children needs clarification. What is the evidence for this specific hypothesis in adults, that justifies re-using the hypothesis that was rejected in children in an adolescent sample? Furthermore, I am not convinced by the choices made in the methods section, that affect the analyses, with the main issue being the index of urbanicity. This warrants revision.

Introduction:

Line 124: “adolescents seem to show different patterns of biological stress reactivity”. Could this be more specific? “different” could be anything. Regarding HR or cortisol levels, I assume there will be concrete evidence in one specific direction.

Lines 130-133: “Results from two neuroimaging studies“ is followed by 3 references. This does not match

Lines 143-147: the reference to the lines feels counter-intuitive. I would suggest to call them A, B and C in order, A resulting in the direct association, and B and C as parts of the indirect way.

Lines 149-150: Here the authors state that the hypothesis was rejected in children. However, the motivation and rationale to investigate the same hypothesis in adolescents may be elaborated on. Why would the factors cited, contributing to vulnerability to mental health problems (lines 152-153) make it plausible that this hypothesis might be true for adolescents? Additionally, comparing it more to the adult literature in light of this specific hypothesis would also validate this choice of hypothesis more.

Line 158, Note to figure 1: the factors mental health and biological stress functioning are well described. Pleas add a definition of urbanicity here as well, to complete the information.

Materials and Methods

Lines 197-198: explaining the procedure of saliva collection, a reference to Figure 2 would be helpful.

Urbanicity: Please state here that the measure investigated is current urbanicity. Further in the discussion this is done, but here should be the place.

To my knowledge, all Dutch studies using CBS data on SAD use a 5 scale division of urbanicity (see for example Van Os, 2002, Frissen 2014/2015 and Peeters 2015). This is also the division given on the CBS website. (https://jeugdmonitor.cbs.nl/begrippen-en-toelichtingen/stedelijkheid-van-een-gebied): “Hierbij zijn vijf categorieën onderscheiden: - zeer sterk stedelijk: gemiddelde oad van 2500 of meer adressen per km2; - sterk stedelijk: gemiddelde oad van 1500 tot 2500 adressen per km2; - matig stedelijk: gemiddelde oad van 1000 tot 1500 adressen per km2; - weinig stedelijk: gemiddelde oad van 500 tot 1000 adressen per km2; - niet stedelijk: gemiddelde oad van minder dan 500 adressen per km2.”

The 5 scale division should be adopted in the manuscript.

Furthermore, in Table 1 urbanicity is given as a mean, but the n per category is much more informative. Were participants equally distributed over the categories, or were they predominantly urban? Since urbanicity is a very important component in this manuscript, it warrants more detailed description and should be comparable to other Dutch sample studies.

And to conclude, in the discussion (lines 533-540) studies of urban upbringing and current city living are compared, as if these measured the same construct. Possibly the different definitions and operationalizations of urbanicity may account for differences in outcome as well.

Lines 267-269: by choosing the extremes of the measurement points (RC1 and 2; RC 3, 4 and 5) it feels like blowing up the differences. I strongly believe that an average would be a more valid value. Please adjust this.

Line 286: T2 n = 1105. But in the participants section, the authors state that T2 data were used (N = 990). And in line 316 509 adolescents participated in the T2 measurement. Somehow these different numbers are confusing and it is not clear on what these differences are based. Please clarify.

Line 324: the participating sample differed from the whole sample in fewer problems, lower neighborhood SES, and older age. The way I read the Table S1 I see MORE self-reported behavioral problems, HIGHER neighborhood SES and older age.

In Lines 297 and further covariates are described. In line 349 it is explained that the analyses are controlled for these variables, but only in the preliminary results it becomes clear which covariate is used in which analysis. Maybe I have missed it, but what is the basis for this decision, and could the specific use be reported earlier?

Discussion:

Lines 477-478: urbanicity is associated with mental health problems in adolescence. I would suggest to be more specific here: only in the behavioral outcome measures such an association was found, and not in the emotional problems. “problems in mental health” therefore seems too strong.

Lines 514-515: urbanicity is associated with acute measures, but not with overall. This is a finding that needs explanation. Is there a specific mechanism that could underlie this relation with acute but not general stress reactivity? I would very much like a bit more discussion than the mere statement about this interesting finding.

Line 584: comorbidity might be too strong a word for a healthy sample with merely subclinical symptoms.

Typo’s:

Line 196: Electrocardio-activity (without a space)

Line 364: “Table” should be removed

Reviewer #3: This interesting ms. asks wehther urbanicity is associated with mental health problems and if this relationship is mediated by biological stress responsivity. The sample consists of 323 adolescents from the Dutch population.

While it is an interesting question to what degree the effects of urbanicity on mental health are mediated by biological stress systems, there are a number of concerns regarding the ms. in its current version, outlined below.

1. The theoretical part and introdcution summarize findings on stress effects on mental health, effects of urbanicity on mental health, and associations with stress biomarkers. The mediation model tested in this ms., however, seems poorly prepared on a conceptual level - why is it that the authors think urbanicity effects on metal health are mediated by stress biomarkers? Why would stress markers not just be a correlate of urbanicity? This needs to be explicated in more detail.

2. Please clarify the use of zip codes. How large is the variation of the population size in a neighbourhood (mean 1400) in the Netherlands? How was the 1km radius implemented? By address or by center of zip code?

3. One theoretical model of urbanicity implies that population density is inversely associated with socioeconomic status. In your mediation analyses, when controlling for SES, the problem arises that you may have controlled for a key mechanism and hence lost the effect of interest. Please also show analyses including SES as a mediator or at least not controlling for SES.

4. Given the high number of correlations under investigation as well as the preselection of variables based on significant associations, how did you control for multiple testing?

5. Please demonstrate how your analyses are powered (e.g. post hoc) for the mediation analyses provided here.

6. PLOS authors have the option to publish the peer review history of their article (what does this mean?). If published, this will include your full peer review and any attached files.

Reviewer #1: Yes: Caio Hofmann Francisco Alves

Reviewer #2: No

Reviewer #3: No

---

## [Author Response · Author response to Decision Letter 0]

11 Dec 2019

Please see the uploaded file: Response to reviewers.

---

## [Decision Letter · Decision Letter 1]

10 Jan 2020

PONE-D-19-18858R1

Urbanicity, biological stress system functioning and mental health in adolescents

PLOS ONE

Dear Dr. Evans,

Thank you for submitting your manuscript to PLOS ONE. After careful consideration, we feel that it has merit but does not fully meet PLOS ONE’s publication criteria as it currently stands. Therefore, we invite you to submit a revised version of the manuscript that addresses the points raised during the review process.

We would appreciate receiving your revised manuscript by Feb 24 2020 11:59PM. To enhance the reproducibility of your results, we recommend that if applicable you deposit your laboratory protocols in protocols.io, where a protocol can be assigned its own identifier (DOI) such that it can be cited independently in the future. For instructions see: http://journals.plos.org/plosone/s/submission-guidelines#loc-laboratory-protocols

We look forward to receiving your revised manuscript.

Kind regards,

Geilson Lima Santana, M.D., Ph.D.

Academic Editor

PLOS ONE

Additional Editor Comments (if provided):

Thank you for addressing the instructions and questions posed by the reviewers. Some minimal details, as indicated by reviewer number 2, need to be tackled before publication.

Best wishes!

Reviewers' comments:

Reviewer's Responses to Questions

**Comments to the Author**

1. If the authors have adequately addressed your comments raised in a previous round of review and you feel that this manuscript is now acceptable for publication, you may indicate that here to bypass the “Comments to the Author” section, enter your conflict of interest statement in the “Confidential to Editor” section, and submit your "Accept" recommendation.

Reviewer #2: All comments have been addressed

Reviewer #3: All comments have been addressed

2. Is the manuscript technically sound, and do the data support the conclusions?

Reviewer #2: Yes

Reviewer #3: Yes

3. Has the statistical analysis been performed appropriately and rigorously? 

Reviewer #2: Yes

Reviewer #3: Yes

4. Have the authors made all data underlying the findings in their manuscript fully available?

Reviewer #2: Yes

Reviewer #3: Yes

5. Is the manuscript presented in an intelligible fashion and written in standard English?

Reviewer #2: Yes

Reviewer #3: Yes

6. Review Comments to the Author

Reviewer #2: I thank the authors for addressing my previous comments in a satisfactory way, and apologise for misreading the urbanicity scale 0-4 = 5 categories.

Following up on this scale, I see in the data, the division between the urban categories a problem many researchers encounter: most participants in the higher urban categories. Since categories 1, 2 and 3 together are almost as big as category 4, I strongly advise to mention the problem of the rural-urban distribution in the limitations section, since it might have affected the data analyses.

And one last minor comment: on the thermometer in S3, I believe in the second step there is missing a black box: twice 0 and then a jump to 2, should be 0, 1, 2, I believe.

Reviewer #3: (No Response)

7. PLOS authors have the option to publish the peer review history of their article (what does this mean?). If published, this will include your full peer review and any attached files.

Reviewer #2: No

Reviewer #3: No

---

## [Author Response · Author response to Decision Letter 1]

13 Jan 2020

Reviewer #2 

1. I thank the authors for addressing my previous comments in a satisfactory way, and apologise for misreading the urbanicity scale 0-4 = 5 categories.

Following up on this scale, I see in the data, the division between the urban categories a problem many researchers encounter: most participants in the higher urban categories. Since categories 1, 2 and 3 together are almost as big as category 4, I strongly advise to mention the problem of the rural-urban distribution in the limitations section, since it might have affected the data analyses.

Response: Thank you for reviewing our revised manuscript. We mention the limitation of uneven distribution of the urbanicity categories in our revision (lines 598-600). The sentence reads: 

Also regarding the neighborhood, our measure of urbanicity was not evenly distributed across the five categories, as most individuals resided in urban or very urban neighborhoods. This uneven distribution may have affected our analyses.

2. And one last minor comment: on the thermometer in S3, I believe in the second step there is missing a black box: twice 0 and then a jump to 2, should be 0, 1, 2, I believe.

Response: Thank you for noticing this. The figure has been revised.

---

## [Editor Report · Decision Letter 2]

22 Jan 2020

Urbanicity, biological stress system functioning and mental health in adolescents

PONE-D-19-18858R2

Dear Dr. Evans,

We are pleased to inform you that your manuscript has been judged scientifically suitable for publication and will be formally accepted for publication once it complies with all outstanding technical requirements.

With kind regards,

Geilson Lima Santana, M.D., Ph.D.

Academic Editor

PLOS ONE

---

## [Editor Report · Acceptance letter]

4 Mar 2020

PONE-D-19-18858R2 

Urbanicity, biological stress system functioning and mental health in adolescents 

Dear Dr. Evans:

I am pleased to inform you that your manuscript has been deemed suitable for publication in PLOS ONE. Congratulations! Your manuscript is now with our production department. 

With kind regards,

on behalf of

Dr. Geilson Lima Santana 

Academic Editor

PLOS ONE